



# Aerodynamic response of a floating wind turbine scale model with inclusion of reference control functionalities

Alessandro Fontanella[1], Elio Daka[1], Felipe Novais[1,2], and Marco Belloli[1]

[1]Mechanical Engineering Department, Politecnico di Milano, Milano, Via La Masa 1, 20156, Italy.
[2]Maritime Research Institute Netherlands (MARIN), Wageningen, 6708 PM, the Netherlands.

**Correspondence:** Alessandro Fontanella (alessandro.fontanella@polimi.it)

**Abstract.** Design and verification of control strategies for floating wind turbines often makes use of aero-hydro-servo-elastic modeling tools. Aerodynamic loads calculation in these tools has been recently validated against experiments not including active wind turbine control. This work investigates the aerodynamic response of a floating wind turbine scale model with active control and platform pitch motion. This is done in wind tunnel testing and with modeling of the scaled system in the offshore

tool OpenFAST. A control design framework is developed to include the reference wind turbine controller ROSCO in the wind tunnel experiment. With platform pitch motion, the turbine aerodynamic response is predicted by the numerical model with different accuracy depending on the turbine control regime. Below rated wind, oscillations of aerodynamic torque in simulations are of lower amplitude than in the experiment, also when dynamic inflow is considered in the aerodynamic model. Above rated wind, where the turbine is controlled with collective blade pitch actuation, the response is not quasi-steady, and

differences between the experiment and simulation are larger than in below-rated wind, in particular for phase with respect to motion.

## 1   Introduction

Floating wind turbine control has been a topic of research since the introduction of floating wind energy. The main reason is the infamous negative damping problem due to the use of the variable-pitch control strategy of bottom-fixed turbines which is

discussed, among many, by Larsen and Hanson (2007) and van der Veen et al. (2012). Most of the research in floating wind turbine control tried to devise new control methodologies to ensure stable operation and reduce fatigue loads for the floating wind turbine components.

Design and verification of control strategies often makes use of aero-hydro-servo-elastic modeling tools to assess the response of the floating system and predict power production. Accuracy of aerodynamic loads calculation in these tools need to

be assessed to ensure correct modeling of the turbine response. This theme has recently been the subject the OC6 Phase III project, which addressed the case of large platform surge and pitch motion in a scaled wind turbine, whose results are presented by Bergua et al. (2022) and by Cioni et al. (2023). Code validation made use of data from the wind tunnel experiment of Fontanella et al. (2021) where no active turbine control strategy was used. The project has shown the aerodynamic response is quasi-static and is correctly captured by codes of different fidelity in case of low-frequency motion and no active turbine





control. Additional verification cases run in the OC6 project, has shown aerodynamic unsteadiness takes place when sinusoidal variation of rotor speed or blade pitch is combined with surge motion, but no experimental data were available to verify codes in this scenario.

In the last decade, several scale model experiments about the wind-wave response of floating wind turbines have been carried out, and a review of them is presented by Gueydon et al. (2020). The large majority of tests involving a scaled wind turbine

did not use active turbine control. One example is the DeepCwind consortium, whose results are summarized by Robertson et al. (2013), that investigated the coupled response of three floating wind turbine concepts, but blade pitch and rotor speed were fixed to a constant value as explained by Goupee et al. (2017). Recently, Mendoza et al. (2022) carried out scale model experiments about a 15 MW floating wind turbine including active control. At the time of writing, wind-only tests with fixed foundation have been examined and used for the validation of three offshore modeling tools. Another research effort in this

topic is the wave basin experiment about a 10 MW floating wind turbine with active control that were carried out by Madsen et al. (2020). Tests with various wind-wave conditions were compared to two offshore codes by Kim et al. (2023); the controller used in the simulation study is the same of the experiment. The code validation study of Kim et al. (2023) addressed the floating wind turbine global response, with simultaneous modeling of multiple uncertain phenomena as hydrodynamic viscous loads, turbulent wind field, closed-loop turbine control, rotor aerodynamics with large motion. Overall, the two codes object of the

validation showed good accuracy. Yu et al. (2017) tested a collective blade-pitch controller system in a wind-wave basin and examined the influence of the turbine controller on the platform global response. A small portion of tests has been modeled with an offshore simulation tool, showing good accuracy for platform motion at the wave frequency. However, in the wave-frequency range, platform response is driven by linear wave excitation and it is difficult to assess the accuracy of aerodynamic loads modeling.

The purpose of this work is to investigate the aerodynamic response of a floating wind turbine scale model with active control and platform pitch motion. This is done in a wind tunnel experiment and with modeling of the scaled system in the offshore tool OpenFAST. Experiments with active control required to develop control tools for the wind turbine scale model. There is no consensus or shared practice on how to implement turbine controls in scale model experiments. Often, the controllers utilized in scale model testing have simplifications with respect to reference controllers used in modeling of utility-scale turbines (e.g.,

the Reference Open Source Controller ROSCO of Abbas et al. (2022b) or the DTU Wind Energy Controller of Meng et al. (2020)). Here, the experiment makes use of the ROSCO, and the same controller of wind tunnel testing is used in simulations. To achieve this goal, a control development framework has been developed and is presented here.

The expected impact of this work is to provide information about the prediction capability of offshore tools with respect to aerodynamic loading in presence of active turbine control. The methodology we developed to integrate active control in

experiments and simulations should benefit future scale model testing activities. Data collected in the experiment, as well as the OpenFAST model of the validation study, are shared with the community to promote studies about control of floating wind turbines. Data and the OpenFAST model can be used for further validation; the ROSCO controller, which has been implemented in Simulink, can be used in future scale model experiments, but also in control studies for utility-scale turbines.



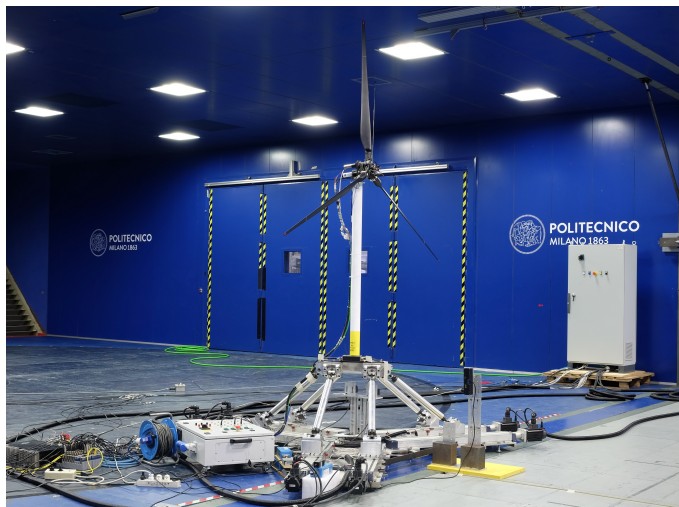
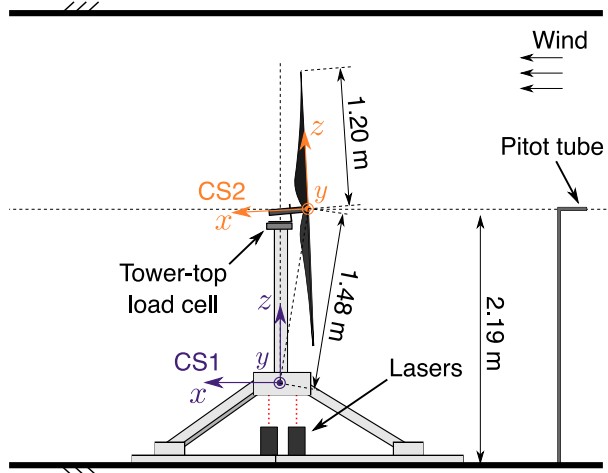

**Figure 1.** Experimental setup in the Polimi wind tunnel. "CS1" and "CS2" are respectively the coordinate systems for platform motion and rotor forces.

## 2 Description of the experimental setup

The experimental setup is shown in Fig. 1. It consists of a wind turbine scale model (WTM), which is mounted on top of a six degrees-of-freedom (DOF) robotic platform. Testing is conducted in the atmospheric boundary layer test section of the Politecnico di Milano wind tunnel, which is 13.84 m wide, 3.84 m high, and 35 m long. The wind turbine is placed 20 m downstream the test section inlet. Tests were performed without roughness elements or turbulence generators for a constant inflow velocity and turbulence intensity of 2%.

### 2.1 Wind turbine aero-servo-dynamic design

The wind turbine of the experiment is a $\lambda_L$ = 1:100 scaled version of the IEA 15 MW, presented by Gaertner et al. (2020). It was designed to reproduce the aerodynamic response of the full-scale turbine with wind speed reduced by a $\lambda_v$ = 1:3.5 factor.

The aim of rotor aerodynamic design is to replicate the blade normal force of the the IEA 15 MW at design tip speed ratio (TSR) of 9 and blade pitch ($\beta$) of 0°. The main difficulty in achieving this goal is Reynolds number, which is 350 times lower

than for the full-scale turbine. The blade design uses the SD7032 airfoil, which has suitable lift and lift-to-drag characteristics at Reynolds number lower than 250k that are expected for the turbine model. The blade chord and twist distributions are altered, section by section, to have the lift force and the variation of lift force with angle-of-attack of the IEA 15 MW.

The wind turbine has active generator control and individual blade pitch control. The wind turbine generator is a brushless DC motor *Maxon EC-4pole-30* with planetary gearbox *Maxon GP32HP* of ratio. Generator speed is measured with the encoder

*ENC 16 EASY* with 500 pulses per turn, and this signal is the feedback for closed-loop control of the turbine scale model. The generator high-speed shaft is connected to the rotor low-speed shaft with a toothed belt of transmission ratio equal to 2; the total



transmission ratio is $\tau_g = 42$ and the transmission efficiency is $\eta_g = 73.5\%$. The electric motor is driven by a *Maxon ESCON 70/10* controller, it functions as a generator with variable torque setpoint computed by the variable-speed control strategy of the turbine controller. The tower is an aluminum tube of 75 mm diameter, and the fore-aft mode is at 9.5 Hz.

The wind turbine has individual blade pitch actuators, housed inside the hub, that are *Harmonic Drive RSF-5B-30-E050-C*. Each pitch actuator is controlled by a *Maxon EPOS 24/2* drive mounted on the turbine hub. Power and blade pitch setpoints are transmitted to individual pitch motors with a 30 channels slip ring. The wind turbine controller computes generator torque and collective blade pitch setpoints for the actuators based on generator speed and wind speed measurements. It runs on a National Instrument PXI embedded control system by means of the Veristand interface.

The main properties of the turbine model are summarized in Table 1.

**Table 1.** Key parameters of the wind turbine model.

| Parameter | Unit | Value |
| --- | --- | --- |
| Rotor diameter | m | 2.400 |
| Blade length | m | 1.110 |
| Hub diameter | m | 0.180 |
| Rotor overhang | m | 0.139 |
| Tilt angle | ° | 5.000 |
| Tower-to-shaft | m | 0.064 |
| Tower diameter | m | 0.075 |
| Tower length | m | 1.400 |
| Nacelle mass | kg | 1.975 |
| Blade mass | kg | 0.240 |
| Rotor mass | kg | 2.041 |
| Rotor inertia | kgm$^2$ | 0.279 |
| Tower mass | kg | 2.190 |


## 2.2   Measurements

Quantities measured in the experiment are rotor forces, platform motion, actual generator speed, collective blade pitch setpoint, hub-height wind speed. Six-component forces at the tower-nacelle interface were measured with an ATI Mini45 load cell with SI-580-20 calibration. Rotor loads are obtained from the projection in the CS2 reference frame of tower-top loads. Platform
pitch motion is measured with two MEL M5L/50 lasers placed beneath the robotic platform. Measurement of the undisturbed wind velocity is obtained with a pitot tube placed at centerline, hub-height, 7.15 m upstream the rotor. Generator speed is measured with the generator encoder and reading of this quantity is an output of the generator drive. Measurement of the





actual blade-pitch angle is not available and is replaced with the collective blade pitch setpoint. All measurements are acquired simultaneously with a NI DAQ with sampling frequency of 2000 Hz.

## 3  Wind turbine control strategy

The wind turbine controller computes generator torque and collective blade pitch setpoints based on generator speed and wind speed measurements. It is uses the algorithms of the ROSCO introduced by Abbas et al. (2022b) and distributed as a DLL and source Fortran code by Abbas et al. (2022a). In this project, the ROSCO has been implemented in MATLAB Simulink, and the same controller implementation is used for the experiment and for co-simulation with OpenFAST.

The logic of the ROSCO used in the experiment is shown in Fig. 2. It consists of two main modules, a generator torque controller that controls generator torque in below-rated wind to achieve maximum wind-power conversion efficiency, and a collective blade pitch controller that controls aerodynamic torque in above-rated wind to limit the extracted power to its nominal value. The generator torque and blade pitch controllers are proportional-integral (PI) controllers with this generic

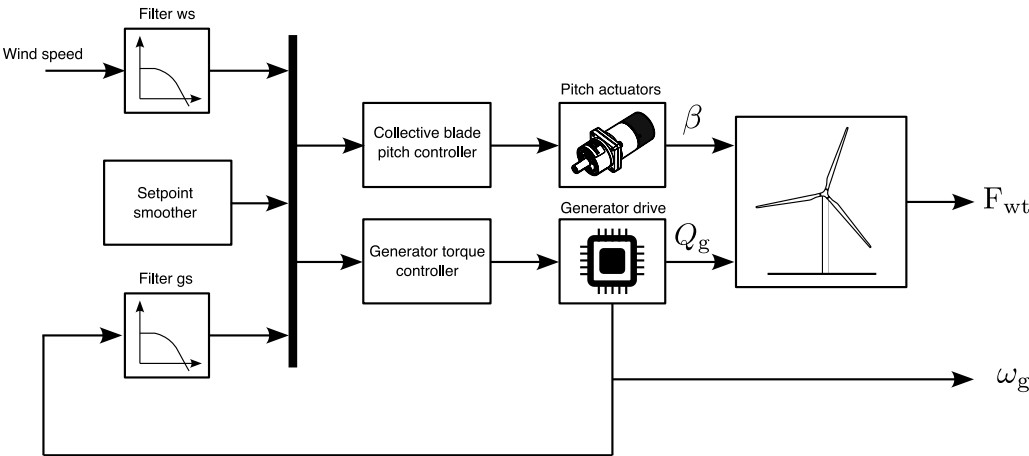

**Figure 2.** Block diagram showing the ROSCO structure and how it is integrated with the wind turbine scale model. The generator torque controller and collective blade pitch controller are based on the PI controller of Eq. 1, "Filter ws" is the low-pass filter for wind speed, "Filter gs" is the low-pass filter for generator speed, $\omega_\mathrm{g}$ is the generator speed signal from the generator encoder, $\mathbf{F}_\mathrm{wt}$ is the 6-components force measured by the tower-top load cell, $Q_\mathrm{g}$ is the generator torque, $\beta$ is the collective blade pitch.

structure:

$$y = k_{\mathrm{P},y}(\omega_{\mathrm{g,s}} - \omega_\mathrm{g}) + k_{\mathrm{I},y}\int_0^T (\omega_{\mathrm{g,s}} - \omega_\mathrm{g})dt,\tag{1}$$

where $y$ is the control input, either generator torque ($y = \mathrm{g}$) of collective blade pitch ($y = \beta$), $k_{\mathrm{P},y}$ and $k_{\mathrm{I},y}$ are the proportional and integral gains, $\omega_\mathrm{g}$ is generator speed, and $\omega_{\mathrm{g,s}}$ is the generator speed set point.



When wind is below rated, blade pitch is held constant to the design value of $0°$, generator torque is controlled to track a constant TSR setpoint $\lambda_0 = 9$ and achieve the maximum power coefficient. In scale model testing, closed-loop TSR tracking is preferred over the more traditional $k\omega^2$ law, also available in ROSCO, because $k\omega^2$ control does not take into account Reynolds-dependency of aerodynamic torque which occurs in small-scale turbines as it is shown by Fontanella et al. (2023a). With TSR tracking:

$$\omega_{\mathrm{g,s}} = \tau_{\mathrm{g}} \frac{\lambda_0 \hat{u}}{R}. \tag{2}$$

where $R$ is rotor radius and $\hat{u}$ is the rotor effective wind speed. In general, this is obtained by means of a wind speed estimator, but in this case it is measured with the hub-height pitot tube upstream the turbine model. Generator speed is filtered with a second-order low-pass filter, wind speed with a first-order low-pass filter.

When the turbine is in above-rated operation, generator torque is held constant:

$$Q_{\mathrm{g}} = \frac{P_0}{\eta_{\mathrm{g}} \tau_{\mathrm{g}} \omega_{\mathrm{r,0}}}, \tag{3}$$

where $P_0$ is the turbine rated power, and $\omega_{\mathrm{r,0}}$ the rated rotor speed. Collective blade pitch is computed with a PI controller as the one of Eq. 1, where the setpoint is $\omega_{\mathrm{g,s}} = \tau_{\mathrm{g}} \omega_{\mathrm{r,0}}$.

In near-rated wind, the setpoint for generator torque and collective blade pitch controllers is the same (i.e., the rated speed). This would lead the controllers to conflict with each other with unwanted oscillations in the turbine response. To avoid this conflict, the setpoint smoothing algorithm of Abbas et al. (2022b) is used, that progressively lowers the generator speed setpoint of one of the two controllers to have smooth transition between one operating regime to the other. Advanced control functionalities available in the ROSCO of Abbas et al. (2022a), such as peak shaving or minimum pitch schedule, are implemented in the MATLB Simulink version of the controller but not used in the wind tunnel study.

### 3.1 Definition of controller parameters

In the experiment, the wind turbine controller is run at model scale. This approach takes a different route with respect to the work of Mendoza et al. (2022), where the turbine controller is run in real-time in its full-scale version with scaling of input and output signals. The full-scale approach does not respect the time scale of the experiment; a proof of this is given in Appendix B.

Parameters of the ROSCO are selected to make it fit the WTM scaling and replicate the static and dynamic response of the IEA 15 MW rotor at model scale. Generator speed setpoints of the generator torque and blade pitch controllers are obtained by downscaling values for the IEA 15 MW and introducing the transmission ratio $\tau_{\mathrm{g}}$. The rated generator torque and the rated generator power are defined to achieve the scaled value of rated rotor torque of the reference turbine (see Eq. 3).

Scaling of gains of the PI generator torque and collective blade pitch controllers does not follow dimensional analysis as the rest of the controller parameters, but seeks to downscale the closed-loop response of the IEA 15 MW given the aerodynamic response and drivetrain parameters of the WTM. Tuning the controller to minimize platform motions is outside the scope of this work, and negative damping of platform modes typically found in floating wind turbines is not an issue here since platform motion is prescribed. The tuning procedure is model-based; dynamics of pitch and generator actuators is neglected. The wind



turbine is modeled as a single degree of freedom system corresponding to the rotor-generator, whose equation of motion is:

$$J^* \dot{\omega}_{\mathrm{r}} = Q_{\mathrm{a}} - \tau_{\mathrm{g}} \eta_{\mathrm{g}} Q_{\mathrm{g}} , \tag{4}$$

where $J^* = J_{\mathrm{r}} + \eta_{\mathrm{g}} \tau_{\mathrm{g}}^2 J_{\mathrm{g}}$ is the total inertia of the rotor and generator, $Q_{\mathrm{a}}$ the rotor aerodynamic torque. With Eq. 4 a dynamic torque balance is imposed at rotor. This equation is used as a basis for control design, as the objective is to reproduce the aero-servo-dynamic response of the rotor rather than the generator. The aerodynamic torque is:

$$Q_{\mathrm{a}} = \frac{1}{2} \rho C_Q(\omega_{\mathrm{r}}, \beta, U) \pi R^3 U^2 , \tag{5}$$

where $\rho$ is air density and $C_Q$ the torque coefficient. $C_Q$ is assumed to be function of rotor speed, collective blade pitch and wind speed $U$, $R$ is rotor radius. The expression of $Q_{\mathrm{a}}$ of Eq. 5 is non linear and is linearized to obtain a linear model of the wind turbine once it is inserted in Eq. 4:

$$Q_{\mathrm{a}} \simeq Q_{\mathrm{a},0} + \left.\frac{\partial Q_{\mathrm{a}}}{\partial \omega_{\mathrm{r}}}\right|_0 (\omega_{\mathrm{r}} - \omega_{\mathrm{r},0}) + \left.\frac{\partial Q_{\mathrm{a}}}{\partial \beta}\right|_0 (\beta - \beta_0) + \left.\frac{\partial Q_{\mathrm{a}}}{\partial U}\right|_0 (U - U_0) , \tag{6}$$

where $(\cdot)_0$ denotes the steady-state value of a quantity for a given turbine operating point. In a more compact form:

$$Q_{\mathrm{a}} \simeq Q_{\mathrm{a},0} + K_{\omega Q} \overline{\omega}_{\mathrm{r}} + K_{\beta Q} \overline{\beta} + K_{UQ} \overline{U} , \tag{7}$$

where $K_{\omega Q}$, $K_{\beta Q}$, $K_{UQ}$ are the aerodynamic torque sensitivities with respect to rotor speed, collective blade pitch and wind speed; $\overline{\omega}_{\mathrm{r}}$, $\overline{\beta}$, $\overline{U}$ are the perturbations of rotor speed, blade pitch and wind speed.

In below-rated operation, blade pitch is fixed ($\overline{\beta} = 0$), wind speed is assumed constant ($\overline{U} = 0$) and combining Eq. 4 with Eq. 1 we have:

$$J^* \ddot{\overline{\theta}}_{\mathrm{r}} - (\tau_{\mathrm{g}}^2 \eta_{\mathrm{g}} k_{\mathrm{P,g}} + K_{\omega Q}) \dot{\overline{\theta}}_{\mathrm{r}} - \tau_{\mathrm{g}}^2 \eta_{\mathrm{g}} k_{\mathrm{I,g}} \overline{\theta}_{\mathrm{r}} = 0 , \tag{8}$$

where $\theta_{\mathrm{r}}$ is rotor azimuth. Gains of the TSR tracking controller are computed from Eq. 8 to have, at model scale, the closed-loop frequency ($\omega_{\mathrm{des},1}$) and damping ($h_{\mathrm{des},1}$) of the IEA 15 MW in below-rated wind:

$$k_{\mathrm{I,g}} = - \frac{J_{\mathrm{sm}}^* \omega_{\mathrm{des},1}^2}{\tau_{\mathrm{g}}^2 \eta_{\mathrm{g}}} , \tag{9}$$

$$k_{\mathrm{P,g}} = - \frac{K_{\omega Q} + 2 J_{\mathrm{sm}}^* \omega_{\mathrm{des},1} \, h_{\mathrm{des},1}}{\tau_{\mathrm{g}}^2 \eta_{\mathrm{g}}} , \tag{10}$$

where $J_{\mathrm{sm}}^*$ is the rotor-generator inertia of the WTM, $\omega_{\mathrm{des},1} = 0.12 \lambda_v \lambda_L^{-1}$ rad/s, and $h_{\mathrm{des},1} = 0.85$. In general, $K_{\omega Q}$ depends on wind speed, but it is about constant in below-rated operation, thus a single $k_{\mathrm{P,g}}$ can be used for any below-rated wind speed.

Above rated, generator torque is constant ($\overline{Q}_{\mathrm{g}} = 0$), and assuming again constant wind speed, Eq. 4 becomes:

$$J^* \ddot{\overline{\theta}}_{\mathrm{r}} + (\tau_{\mathrm{g}} K_{\beta Q} K_{\mathrm{P},\beta} - K_{\omega Q}) \dot{\overline{\theta}}_{\mathrm{r}} + \tau_{\mathrm{g}} K_{\beta Q} K_{\mathrm{I},\beta} \overline{\theta}_{\mathrm{r}} = 0 . \tag{11}$$





In above-rated wind, aerodynamic sensitivities $K_{\omega Q}$ and $K_{\beta Q}$ depend on wind speed. Gains of the collective pitch controller are computed for discrete wind speeds from rated to cut-out to have for the WTM the same response of the IEA 15 MW. The closed-loop frequency and damping of the IEA 15 MW at wind speed $U_0$ are:

$$\omega_{\text{des},2}(U_0) = \sqrt{\frac{\tau_g K_{\beta Q,\text{fs}}(U_0)\, k_{\text{I},\beta,\text{fs}}}{J_{\text{fs}}^*}}, \tag{12}$$


$$h_{\text{des},2}(U_0) = \frac{\tau_g K_{\beta Q,\text{fs}}(U_0) k_{\text{P},\beta,\text{fs}} + K_{\omega Q,\text{fs}}(U_0)}{2 J_{\text{fs}}^*\, \omega_{\text{des},2}(U_0)}, \tag{13}$$

where $(\cdot)_{\text{fs}}$ denotes full-scale quantities. Gains for the WTM at that wind speed are:

$$k_{\text{I},\beta}(U_0\lambda_v) = \frac{J_{\text{sm}}^*\, (\omega_{\text{des},2}(U_0)\, \lambda_v\lambda_L^{-1})^2}{\tau_g K_{\beta Q,\text{sm}}(U_0\lambda_v)}, \tag{14}$$

$\quad k_{\text{P},\beta}(U_0\lambda_v) = \dfrac{K_{\omega Q,\text{sm}}(U_0\lambda_v) + 2 J_{\text{sm}}^*\, (\omega_{\text{des},2}(U_0)\, \lambda_v\lambda_L^{-1})\, h_{\text{des},2}(U_0)}{\tau_g K_{\beta Q,\text{sm}}(U_0\lambda_v)},$ $\qquad(15)$

where $(\cdot)_{\text{sm}}$ denotes quantities for the scaled turbine. Equations 14-15 result in the gain schedule for the pitch controller, where the scheduling variable is wind speed. However, since at each wind speed corresponds a steady-state value of collective blade pitch, this is used as the scheduling variable. In the WTM there is no feedback of the actual pitch angle and it is replaced with the pitch angle setpoint at previous time step.

Calculation of gains by means of Eq. 9-10 and Eq. 14-15 requires to compute the aerodynamic sensitivities of the wind turbine at the steady-state operating points. Steady-state conditions of the IEA 15 MW (shown in Fig. 5) were used as a first guess to compute the aerodynamic sensitivities required fro controller tuning based on the $C_P(\lambda,\beta)$ table of the WTM OpenFAST model, and obtain reasonable values of PI gains. Then, with these gains, the WTM OpenFAST model was simulated to obtain a new set of steady-state points, that were used to refine the controller tuning.

Gains of the generator torque controller for the WTM are $k_{\text{P},g} = -8.3 \times 10^{-3}$ Nm/rad/s, instead of $-1.1 \times 10^{-1}$ Nm/rad/s for the scaled IEA 15 MW, and $k_{\text{I},g} = -1.9 \times 10^{-2}$ Nm/rad instead of $-3.7 \times 10^{-1}$ Nm/rad. Pitch controller gains for the WTM are compared at model scale with those of the IEA 15 MW in Fig.3. Gains for the WTM are significantly different than values obtained by scaling gains of the IEA 15 MW, the main reason for this difference being rotor inertia, which is larger for the scale model ($0.279$ kgm$^2$) compared to the scaled value of the IEA 15 MW ($0.031$ kgm$^2$); aerodynamic sensitivities

for the WTM are instead close to the reference wind turbine (see Fig. 7). Interestingly, when rotor inertia and aerodynamic sensitivities of the turbine scale model are ideally downscaled the controller tuning procedure is equivalent to scaling the gains of the full-scale turbine.

## 4    Methodology for investigation of the turbine response

The wind turbine is subjected to prescribed platform pitch motion, aerodynamic loads are calculated from tower-top load mea-

surements and are compared to two models of the scaled turbine, one in OpenFAST and one based on linearized aerodynamics.



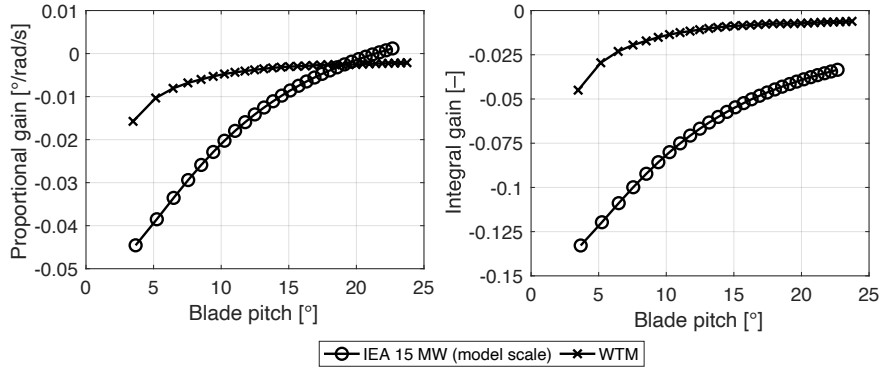

**Figure 3.** Proportional and integral gains for the collective blade pitch controller of the IEA 15 MW (at model scale) and the wind turbine scale model (WTM) as function of collective blade pitch, which is used for scheduling.

This section summarizes the platform motion conditions of the experiment, it explains the algorithm we used to estimate aerodynamic loads from tower-top forces, it provides a description of the modeling approach adopted in OpenFAST, and derives equations of the linearized aerodynamic model.

## 4.1 Wind turbine operating conditions and platform motion

The wind turbine in fixed configuration is run at several wind speeds ranging from 2.5 m/s to 5.8 m/s to measure the steady-state response of rotor torque and thrust, rotor speed and collective blade pitch.

Two functioning conditions are selected for tests with platform motion, corresponding to wind speeds of 2.87 m/s and 5.05 m/s. The imposed motion emulates large-amplitude platform tilt oscillations in floating wind turbines. Motion is in pitch direction, i.e., rotation about the y-axis of the CS1 reference frame (see Fig. 1) and is sinusoidal:

$$\theta(t) = A_\mathrm{m} \sin(2\pi f_\mathrm{m} t), \tag{16}$$

where $A_\mathrm{m}$ is motion amplitude and $f_\mathrm{m}$ is motion frequency. Rotor-level unsteadiness due to the global response of the rotor and its wake, is often associated to the rotor reduced frequency $f_\mathrm{r}$ defined as:

$$f_\mathrm{r} = \frac{f_\mathrm{m} D}{U_0}, \tag{17}$$

where $D$ is rotor diameter. Several combinations of $A_\mathrm{m}$ and $f_\mathrm{m}$ are run in the experiment to explore the turbine aerodynamic
response at various $f_\mathrm{r}$ and with different amplitude of apparent wind speed oscillations $\Delta U = 2\pi f_\mathrm{m} A_\mathrm{m}$. Motions conditions are summarized in Table 2.

## 4.2 Calculation of rotor aerodynamic loads

Aerodynamic rotor thrust and torque are obtained from measurements of tower-top interface forces. Measurements are processed to remove the force contribution due to inertia of the rotor-nacelle assembly, which is subjected to an acceleration when



**Table 2.** Motion conditions ($A_\mathrm{m}$ is amplitude of pitch motion, $f_\mathrm{m}$ is frequency, $\Delta U$ is the apparent wind speed at hub-height, $f_\mathrm{r}$ BR is the reduced frequency with below rated wind of 2.87 m/s, $f_\mathrm{r}$ AR the reduced frequency with above rated wind of 5.05 m/s).

| $A_\mathrm{m}$ [°] | $f_\mathrm{m}$ [Hz] | $\Delta U$ [m/s] | $f_\mathrm{r}$ BR [−] | $f_\mathrm{r}$ AR [−] |
|---|---|---|---|---|
| 3.2 | 0.25 | 0.13 | 0.21 | 0.12 |
| 2.2 | 0.25 | 0.09 | 0.21 | 0.12 |
| 1.1 | 0.25 | 0.04 | 0.21 | 0.12 |
| 3.3 | 0.75 | 0.40 | 0.63 | 0.36 |
| 2.2 | 0.75 | 0.26 | 0.63 | 0.36 |
| 1.1 | 0.75 | 0.13 | 0.63 | 0.36 |
| 2.2 | 1.25 | 0.45 | 1.05 | 0.59 |
| 1.7 | 1.25 | 0.34 | 1.05 | 0.59 |
| 1.1 | 1.25 | 0.22 | 1.05 | 0.59 |
| 2.0 | 1.50 | 0.48 | 1.25 | 0.71 |
| 1.7 | 1.50 | 0.41 | 1.25 | 0.71 |
| 1.1 | 1.50 | 0.27 | 1.25 | 0.71 |
| 1.1 | 1.75 | 0.31 | 1.46 | 0.83 |
| 0.8 | 1.75 | 0.23 | 1.46 | 0.83 |
| 0.5 | 1.75 | 0.15 | 1.46 | 0.83 |
| 0.8 | 2.00 | 0.27 | 1.67 | 0.95 |
| 0.5 | 2.00 | 0.18 | 1.67 | 0.95 |
| 0.2 | 2.00 | 0.07 | 1.67 | 0.95 |

platform moves. For every motion condition, two tests are run where the same type of motion (amplitude and frequency) is prescribed to the wind turbine; in one test there is no wind, the rotor is fixed, and loads measured by the load cell are mostly due to inertia (i.e., the contribution due to air drag on the turbine components is negligible); in the test with wind, the load cell measures inertia and aerodynamic forces. Time series acquired in the two tests are windowed so they have the same integer number of motion periods; time series of forces in the test with no wind are subtracted from time series of forces in the test

with wind, after being projected from CS1 to CS2.

Rotor speed is regulated by the wind turbine controller and in general, when the turbine operates in unsteady conditions, rotor speed is not constant. The inertia torque due to rotor acceleration is present in the load cell measurements. With sinusoidal platform motion, rotor speed oscillations are dominated by the harmonic component at the motion frequency, as it is shown in Fig. 4. Rotor speed oscillations at frequencies other than $f_m$ are regarded as noise. Aerodynamic torque is computed removing

the torque component due to rotor inertia from $M_x$ of CS2:

$$Q(t) = M_x(t) - J_\mathrm{r} A_\omega (2\pi f_\mathrm{m})^2 \cos(2\pi f_\mathrm{m} t + \phi_\omega), \tag{18}$$



where $A_\omega$ and $\phi_\omega$ are the amplitude and phase of the rotor speed spectrum.

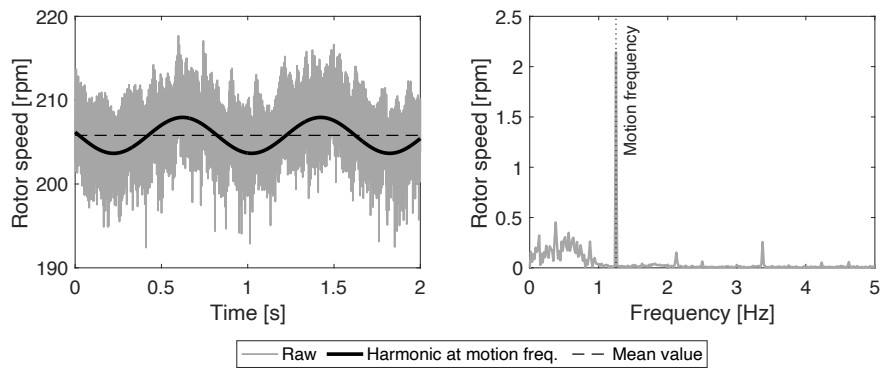

**Figure 4.** Rotor speed with prescribed platform pitch motion of $f_\mathrm{m}$ = 1.25 Hz, $A_\mathrm{m}$ = 2.2°. Left: time series, right: spectrum.

### 4.3 OpenFAST model

An aero-servo-elastic model of the wind turbine scale model of the experiment is created in OpenFAST (v3.1.0). The wind
turbine is simulated at full-scale. This is done to avoid the use of small time steps, and because mapping of aerodynamic
loads to the structural module of OpenFAST has been shown to be inaccurate with small amplitude forces of scale model rotors
(forces for the wind turbine scale model are 122500 time smaller than for the IEA 15 MW). Accuracy of this mapping is needed
to simulate the wind turbine controller. Blades and tower are modeled as rigid bodies. A damped oscillator is introduced at
the base of the wind turbine and external forces are applied to it to prescribe the platform motion recorded in the experiment.
The aerodynamic model is implemented in AeroDyn15 based on blade twist and chord radial distributions, and Reynolds-
dependent polars at 39 radial stations. Calculation of induced velocity in the BEM model of AeroDyn can be based on wake
equilibrium assumption (steady BEM, SB) or on dynamic wake (dynamic BEM, DB); the airfoil model can be based on static
polars (SA), or account for flow hysteresis during attached flow and dynamic stall (unsteady airfoil, UA). Four combinations
of these modeling approaches are considered for simulation of the experiment.

The wind turbine controller is the same Simulink controller of the experiment which is run in co-simulation with the Open-
FAST model. Parameters are upscaled from those used in scaled testing by means of dimensional analysis to be consistent with
the rest of the OpenFAST model. Pitch actuators are modeled as third-order systems of transfer function:

$$G_\mathrm{act}(s) = \frac{b_1 s^2 + b_2 s + b_3}{a_1 s^3 + a_2 s^2 + a_3 s + a_4} . \tag{19}$$

Coefficients of $G_\mathrm{act}(s)$ are obtained by means of system identification carried out on the WTM before wind tunnel testing.
In the frequency range of imposed motion tests, $G_\mathrm{act}(s)$ introduces a constant time delay of 0.075 s (i.e., phase is linear with
frequency) and unit amplification.



#### 4.4 Linearized model of rotor aerodynamic loads

Aerodynamic rotor thrust is written with the same formulation used for torque in Eq. 5 as:

$$T = \frac{1}{2}\rho C_T(\omega_{\mathrm{r}}, \beta, U)\pi R^2 U^2 , \tag{20}$$

where $C_T$ is the thrust coefficient. Equation 20 is linearized with the same approach used for Eq. 5 and becomes:

$$T \simeq T_0 + K_{\omega T}\overline{\omega_{\mathrm{r}}} + K_{\beta,T}\overline{\beta} + K_{UQ}\overline{U} . \tag{21}$$

In case of a floating wind turbine and steady wind, $\overline{U} = -\dot{x}_{\mathrm{hub}}$ is the apparent wind speed for the rotor due to rigid-body platform motion, where $\dot{x}_{\mathrm{hub}}$ is the hub velocity normal to the rotor plane. Equations 21-6 become:

$$T \simeq T_0 + K_{\omega T}\overline{\omega_r} + K_{\beta T}\overline{\beta} - K_{UT}\dot{x}_{\mathrm{h}} , \tag{22}$$

$$Q \simeq Q_0 + K_{\omega Q}\overline{\omega_r} + K_{\beta Q}\overline{\beta} - K_{UQ}\dot{x}_{\mathrm{h}} . \tag{23}$$

The linearized model of Eq. 22-23 is used to verify if the aerodynamic response with platform pitch motion and active control follows the quasi-steady theory. If quasi-steady theory is valid, the total variation of aerodynamic loads is the sum of variations induced by apparent wind, rotor speed and blade pitch oscillations. With this assumption, the thrust and torque oscillations due to apparent wind is computed from total aerodynamic loads as:

$$\Delta T = (F_x - T_0) - (K_{\omega T}\overline{\omega_r} + K_{\beta T}\overline{\beta}) , \tag{24}$$

$$\Delta Q = (Q - Q_0) - (K_{\omega Q}\overline{\omega_r} + K_{\beta Q}\overline{\beta}) , \tag{25}$$

where $F_x$ and $M_x$ are the aerodynamic force and torque in the x-axis of CS2, respectively, and obtained with the algorithm of Sect. 4.2. Platform pitch motion results in rotor speed oscillations and blade pitch actuations with the same frequency of motion:

$$\overline{\omega_{\mathrm{r}}}(t) = A_\omega \cos(2\pi f_{\mathrm{m}}t + \phi_\omega) , \tag{26}$$

$$\overline{\beta}(t) = A_\beta \cos(2\pi f_{\mathrm{m}}t + \phi_\beta) , \tag{27}$$

where $A_\beta$ and $\phi_\beta$ are the amplitude and phase of the spectrum of blade pitch $\beta$, evaluated at frequency equal to $f_{\mathrm{m}}$. For OpenFAST results, $\beta$ is the actual value of blade pitch available among simulation outputs; for experimental results $\beta$ is





obtained from the convolution of the collective blade pitch setpoint and the pitch actuator transfer function of Eq. 19. With harmonic motion, the variation of thrust force and torque due to apparent wind is:

$$\Delta T(t) = -K_{UT}(2\pi f_{\mathrm{m}})A_{\mathrm{hub}}\cos(2\pi f_{\mathrm{m}}t)\,, \tag{28}$$


$$\Delta Q(t) = -K_{UQ}(2\pi f_{\mathrm{m}})A_{\mathrm{hub}}\cos(2\pi f_{\mathrm{m}}t)\,, \tag{29}$$

with $A_{\mathrm{hub}} = A_{\mathrm{m}}d_{\mathrm{hub}}$, where $d_{\mathrm{hub}}$ is the hub distance from the center of platform pitch rotation ($d_{\mathrm{hub}}$ = 1.48 m, see Fig. 1). When the turbine aerodynamic response is described by Eq. 28-29, the phase of the force response with respect to motion is $-\pi/2$, and the zero-peak amplitude normalized by amplitude of hub motion is linear with motion frequency:

$$\Delta T/A_{\mathrm{hub}} = 2\pi f_{\mathrm{m}}K_{UT}\,, \tag{30}$$

$$\Delta Q/A_{\mathrm{hub}} = 2\pi f_{\mathrm{m}}K_{UQ}\,. \tag{31}$$

## 5 Results

This section presents results about the wind turbine response from experimental measurements and OpenFAST simulations.
First, the rotor speed-blade pitch-thrust-torque characteristics with fixed turbine controlled with the ROSCO are presented. Next, the rotor performance coefficients obtained with steady wind and several combinations of fixed TSR-blade pitch are discussed and utilized to compute the sensitivities of the linearized aerodynamic model of Sect. 4.4. Finally, the wind turbine response with platform pitch motion and closed-loop control is analyzed; phase-averaged time series of rotor speed, blade pitch, aerodynamic thrust and torque are examined to validate the capability of OpenFAST to predict the turbine behavior;
aerodynamic thrust and torque response to apparent wind is computed by means of the linearized model to assess the presence of unsteadiness.

### 5.1 Fixed turbine response

The response of the WTM controlled with the ROSCO is measured at six wind speeds. Figure 5 shows the operating points obtained in the wind tunnel, which are compared at model scale to curves computed in OpenFAST for the IEA 15 MW and
for the wind turbine scale model. The rotor speed characteristic of the wind turbine scale model measured in the wind tunnel matches with good accuracy the IEA 15 MW The OpenFAST model of the WTM is perfectly overlapping with the reference, whereas rotor speed in the experiment is slightly higher (the maximum error is 10.5 rpm at 2.9 m/s). The discrepancy can be due to a small static offset in the generator speed feedback or in the wind speed measurement used for calculation of TSR and the generator speed setpoint. In below-rated wind, collective blade pitch is 2.3° instead of 0°, and this deviation from the



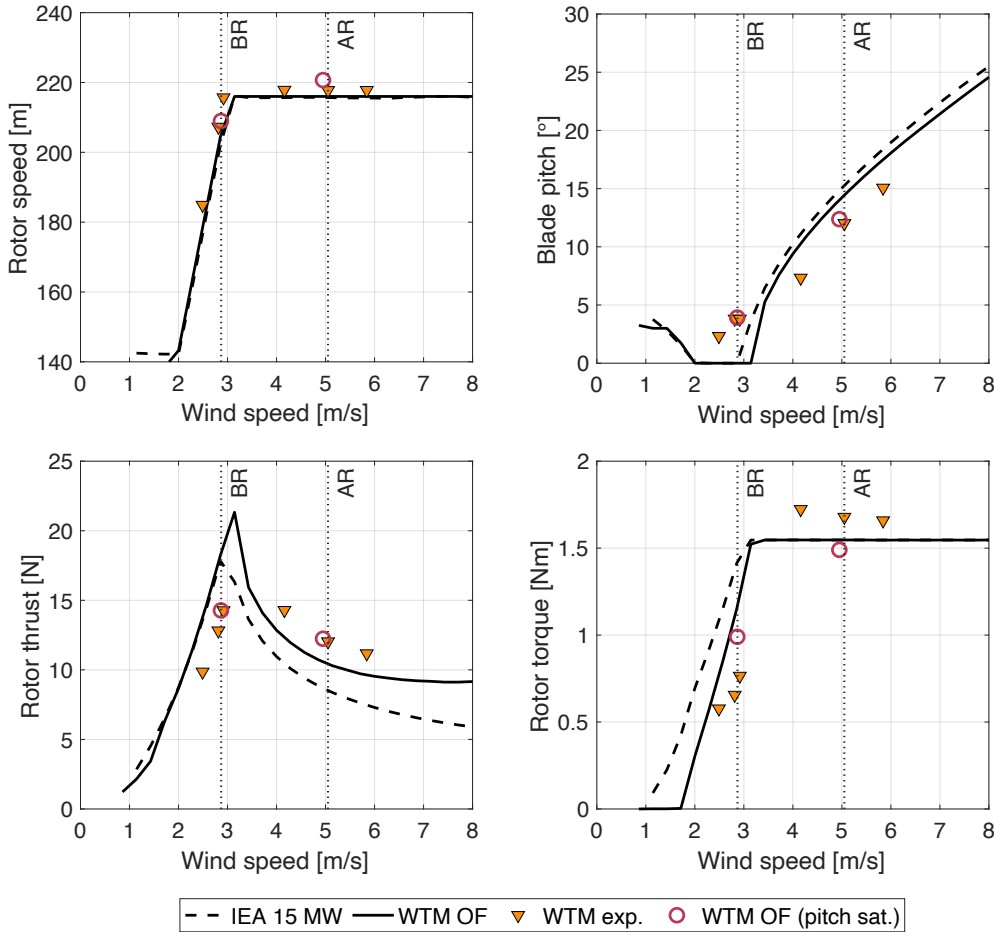

**Figure 5.** Steady-state operating points for the wind turbine scale model (WTM) obtained in the wind tunnel experiment (exp.) and from the OpenFAST model (OF) are compared to the IEA 15 MW at model scale. Vertical dotted lines identify the below rated (BR) and above rated (AR) operating conditions considered for tests with platform motion. In "WTM OF (pitch sat.)" the minimum pitch in ROSCO is set to the BR pitch of the experiment to simulate the blade pitch offset.

reference is likely due to misalignment of individual blades and/or an incorrect setting of the zero-pitch position. In above-rated wind, the rated rotor speed is achieved with values of blade pitch that have an offset of about -3.5° with respect to the IEA 15 MW. The scale model rotor is designed to match the thrust force of the IEA 15 MW in below rated wind, when TSR = 9 and $\beta = 0°$, and this is true for the OpenFAST model of the WTM where target values of rotor speed and blade pitch are achieved. Correct scaling of rotor torque is not the primary objective of blade design, and the scale model torque predicted

by OpenFAST is lower than target for any wind speed: this is due to the lower efficiency of the SD7032 compared to airfoils of the IEA 15 MW. In below-rated wind, the WTM in the experiment works with TSR and collective pitch slightly different than those considered for rotor design, which results in a decreased in thrust and torque. In OpenFAST simulations, the turbine





scale model achieves the rated rotor speed and rotor torque with pitch angle values slightly lower than the IEA 15 MW; thrust force is higher than target due to the aerodynamic characteristics of the airfoil used in the scale model blade. In the experiment,

torque is higher than the rated value for the IEA 15 MW; the cause of this error can be the torque setpoint obtained with Eq. 3, which requires knowledge of the transmission efficiency efficiency. Rotor thrust has the same trend in the experiment and in the WTM OpenFAST model, but wind tunnel values are higher than in simulations. The difference is attributed to values of blade pitch that are lower than target.

The minimum blade pitch of ROSCO in the WTM OpenFAST model is increased to the collective pitch of the experiment to

simulate the blades offset. This model is simulated in the two wind conditions considered in the tests with platform movement and results are closer to those of the experiment. The largest difference is seen for rotor torque, which is 29% higher in OpenFAST than in the experiment.

### 5.2 Performance coefficients and linearized aerodynamic response

Power and thrust coefficients of the wind turbine scale model are measured for various combinations of TSR and blade pitch.

Wind speed is 4 m/s in all tests and TSR is varied changing rotor speed in open-loop (i.e, without using the ROSCO). The same conditions of the experiment are simulated with the OpenFAST model of the WTM, and results of wind tunnel measurements and simulations are compared in Fig. 5. The shape of $C_P$ and $C_T$ calculated in OpenFAST is very close to those measured in the wind tunnel. The maximum $C_P$ is 0.44 for (TSR $\approx 9.5$, $\beta = 0°$); the transition from maximum to zero is milder in the experiment than in OpenFAST. Numerical and experimental $C_T$ are similar.

Aerodynamic sensitivities are calculated based on the expressions reported in Appendix A, based on partial derivatives of the performance coefficients of Fig. 6 and the steady-state operating points of Fig. 5. Partial derivatives of $C_P(\lambda, \beta)$ and $C_T(\lambda, \beta)$ are obtained from the numerical gradient of rotor performance coefficients which is computed with the central difference method. Sensitivities of the IEA 15 MW are compute with the same procedure used for the WTM. The aerodynamic sensitivities of the WTM are shown in Fig. 7, together with those of the IEA 15 MW. In above-rated wind, the experiment is in

good agreement with the OpenFAST model of the WTM and match the IEA 15 MW, whereas larger differences are seen below rated. Discrepancies in partial regime are more pronounced for $K_{\beta Q}$ and $K_{\beta T}$, the sensitivities to blade pitch, and can be due to the steady-state pitch, which is different from $0°$ in the experiment. $K_{\omega Q}$ is very similar in the experiment and in OpenFAST, and this supports the use of the OpenFAST results in the tuning of the PI TSR-tracking controller; a similar convergence of results is found for $K_{\beta Q}$ in above-rated wind, thus the OpenFAST model is suitable also for design of the PI pitch controller.

Aerodynamic sensitivities of Fig. 7 constitute the basis of the linearized aerodynamic model of Sect. 4.4. In general, the linearized aerodynamic response of the experimental and OpenFAST is expected to be similar where convergence of aerodynamic sensitivities is achieved. Dissimilarity of $K_{\beta Q}$ and $K_{\beta T}$ in below-rated wind do not impact the turbine response, because pitch is not actuated at that wind speeds. The turbine exhibits different sensitivity of torque to wind speed ($K_{UQ}$) in the below-rated region, and of thrust to blade pitch ($K_{\beta T}$) in above-rated regime.





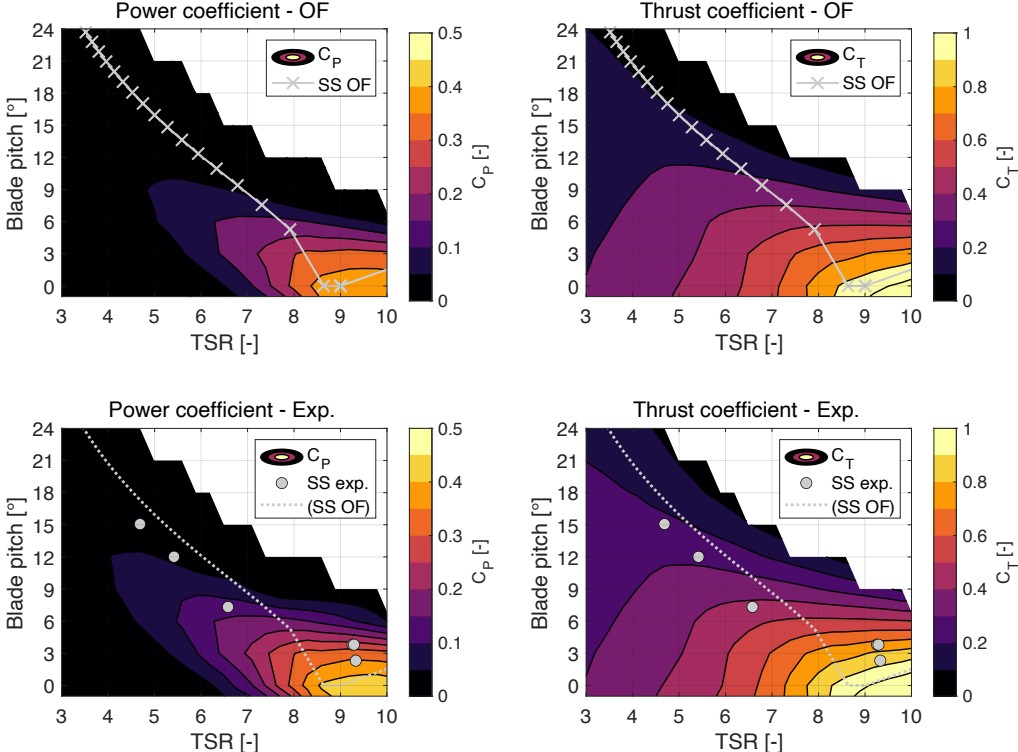

**Figure 6.** Power coefficient ($C_P$) and thrust coefficient ($C_T$) surfaces for the wind turbine scale model measured in the wind tunnel (Exp.) and obtained from OpenFAST (OF). "SS OF" and "SS exp." are the steady-state control trajectories of Fig. 5. Negative values are not shown.

## 5.3 Response with platform pitch motion

The wind turbine is subjected to prescribed platform pitch motion of frequency and amplitude reported in Table 2. In the dynamic wind condition created by platform movement the turbine controller actuates generator torque and collective blade pitch to regulate rotor speed. Platform pitch variations result in oscillations of rotor speed, blade pitch, rotor thrust and torque. The turbine response recorded over a number of periods is phase-averaged using the platform pitch motion for the synchronizing signal, in order to filter harmonic contributions that are not due to platform motion.

Figure 8 shows the WTM response with below-rated wind and three motion conditions. Blade pitch is saturated and the controller responds with actuation of generator torque. The minimum blade pitch in OpenFAST is set to the same value of the experiment to simulate the blade pitch offset. Blade pitch is slightly higher than in the fixed case, and this results in slightly lower mean value for rotor speed, rotor thrust and aerodynamic torque. The other signals exhibit a first-order sine wave, thus the wind turbine response is driven by a single frequency corresponding to platform motion. Rotor speed has oscillations of few rpm, the mean value is lower than the steady-state value of Fig. 5, but it is similar in OpenFAST and in the experiment; the amplitude of oscillations is the largest with $f_\mathrm{m} = 1.25$ Hz, and minimum with $f_\mathrm{m} = 0.15$ Hz, thus it appears to be proportional





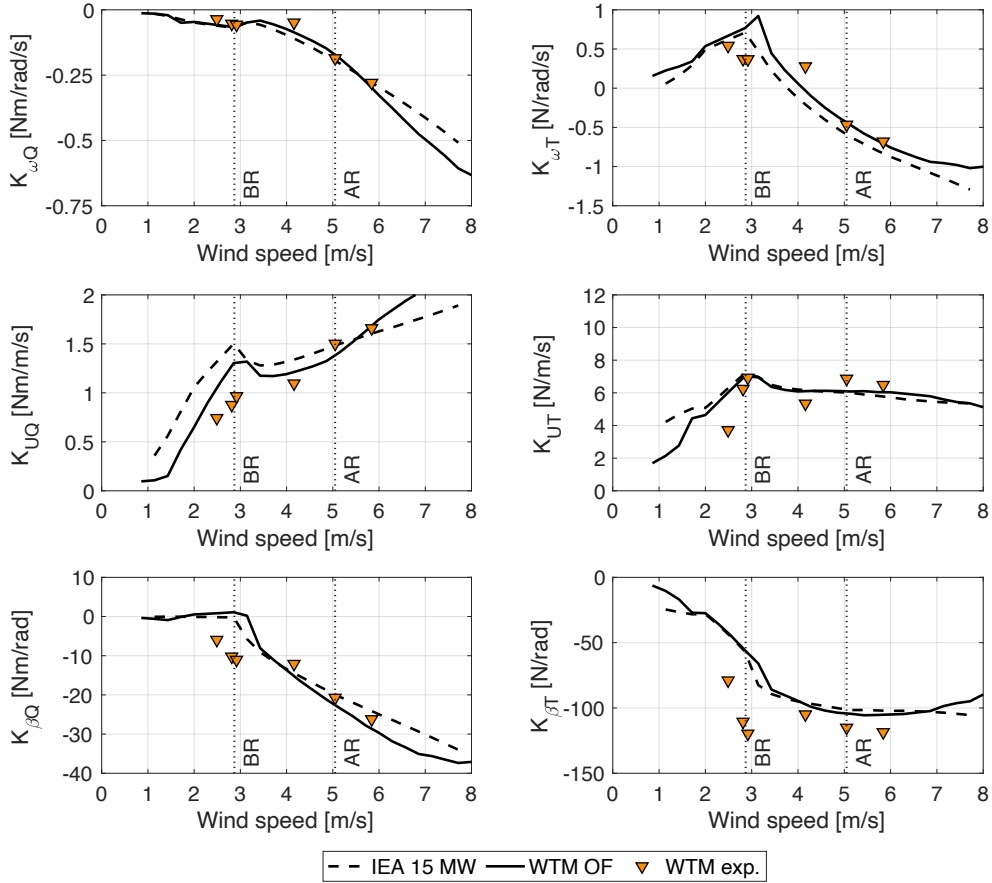

**Figure 7.** Aerodynamic sensitivities at the steady-sate operating points of Fig. 6 obtained from $C_P(\lambda, \beta)$ and $C_T(\lambda, \beta)$ coefficients. Experimental results (WTM exp.) are compared to values computed from the OpenFAST model of the turbine scaled model (WTM OF) and to the IEA 15 MW at model scale. Vertical dotted lines identify the below rated (BR) and above rated (AR) operating conditions considered for tests with platform motion.

to the apparent wind speed created by platform motion ($\Delta U$ in Table 2); the amplitude is larger in the experiment than in OpenFAST, and the phase with respect to motion is different.

Aerodynamic thrust and torque have a maximum with motion phase of $\approx 180°$, that corresponds to the hub moving upwind with maximum velocity; thus the aerodynamic response of the wind turbine is driven by apparent wind created by motion rater than rotor speed oscillations. Peak thrust is slightly higher in OpenFAST than in the experiment and small higher-order effects are seen with $f_m = 2$ Hz, but the overall agreement is good. Differences are larger for torque. Torque oscillations have lower amplitude in OpenFAST than in the experiment, regardless of the output variable; in cases with $f_m = 1.25$ Hz and 2.00 Hz the peak is at $\approx 180°$ in OpenFAST, and at $\approx 170°$ in the experiment. Torque in OpenFAST is also computed from tower-top



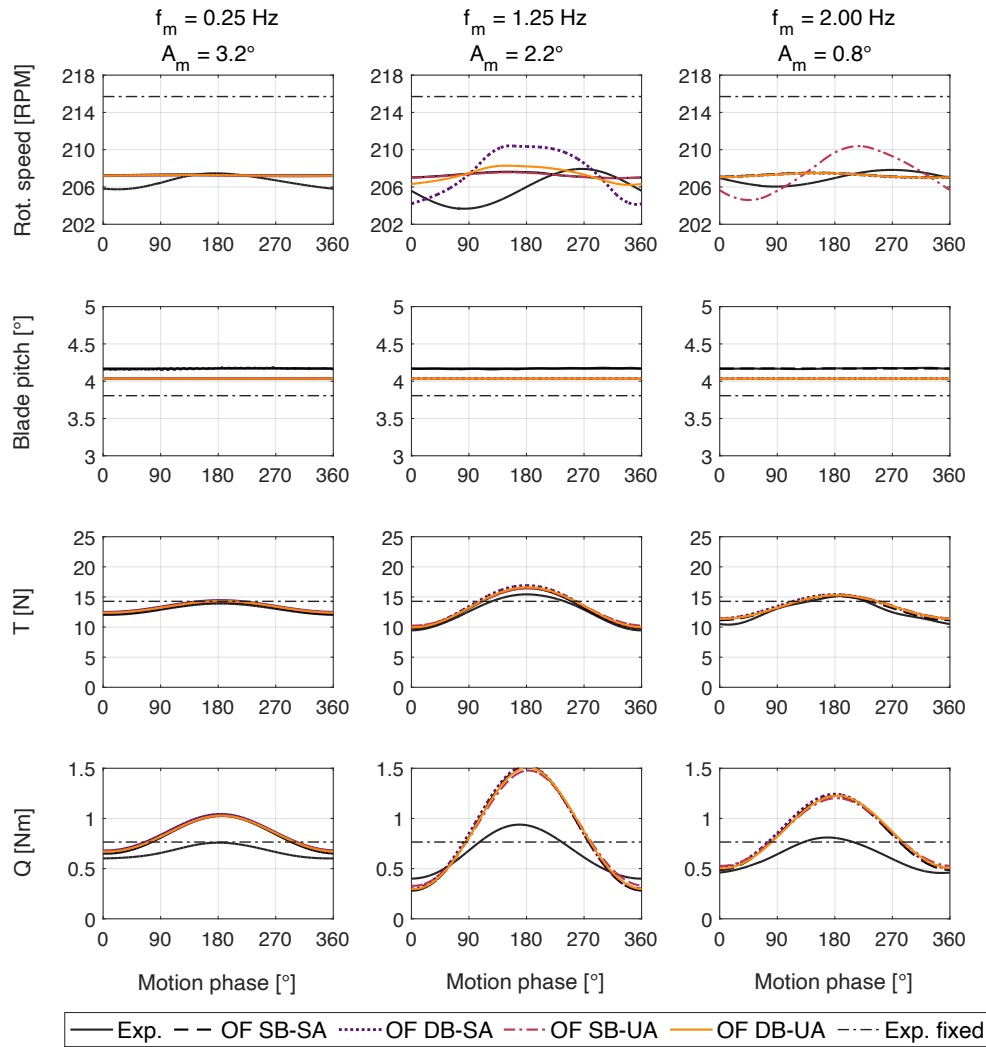

**Figure 8.** Phase-averaged turbine response ($T$ is aerodynamic rotor thrust, $Q$ is aerodynamic torque) with platform pitch motion of various frequency ($f_m$) in below-rated wind, measured in the experiment (Exp.) and obtained in OpenFAST (OF) simulations (SB = steady BEM, DB = dynamic BEM, SA = static airfoil polars, UA = unsteady airfoil polars). "Exp. fixed" is the steady-state value with fixed turbine.

loads with the algorithm utilized for experimental data (see Sect. 4.2) and it is similar to the aerodynamic torque computed in AeroDyn; this result is not reported in Fig. 9, but confirms the post process of experimental data is correct.

OpenFAST solutions with unsteady airfoil polars are very close to those with static polars. The amplitude and frequency of motion ensures dynamic stall is confined to blade root and hysteresis in airfoil aerodynamic response is negligible due to low frequency of motion (Sebastian and Lackner (2013)). The amplitude of thrust and torque oscillations in OpenFAST is slightly higher with dynamic BEM than with steady BEM as it has been observed by Bergua et al. (2022), for harmonic rotor speed






with amplitude of 15% the mean value. However, the amplitude of experimental aerodynamic torque is not matched by any OpenFAST model.

Figure 9 shows the phase-averaged response with above-rated wind and three motion conditions. In this case, generator

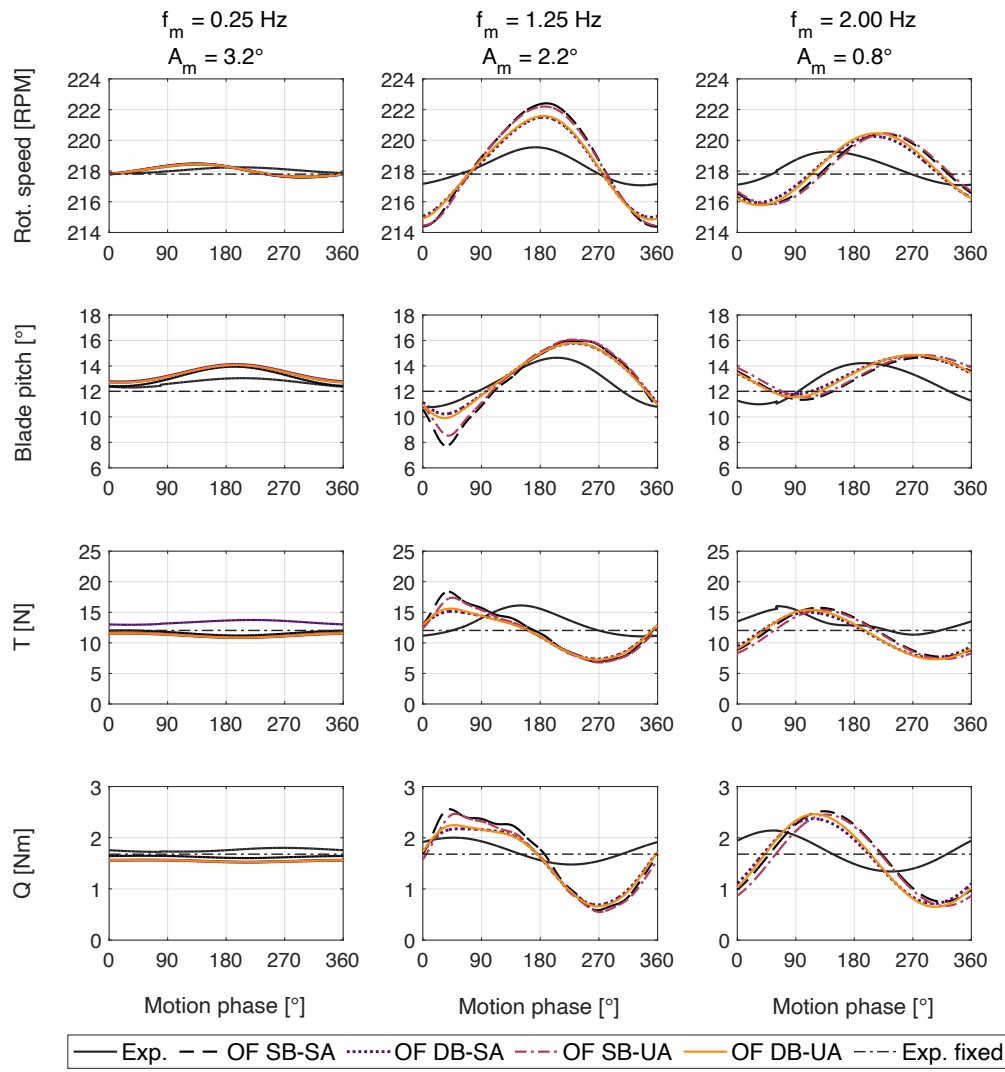

**Figure 9.** Phase-averaged turbine response ($T$ is aerodynamic rotor thrust, $Q$ is aerodynamic torque) with platform pitch motion of various frequency ($f_{\mathrm{m}}$) in above-rated wind, measured in the experiment (Exp.) and obtained in OpenFAST (OF) simulations (SB = steady BEM, DB = dynamic BEM, SA = static airfoil polars, UA = unsteady airfoil polars). "Exp. fixed" is the steady-state value with fixed turbine.

torque is saturated to its rated value, and the controller responds with actuation of collective blade pitch. The mean value of all signals is aligned with the fixed turbine case, and it is similar in the experiment and OpenFAST. Signals generally exhibit a first-order sine wave, but some higher-order effects are seen in the case with $f_{\mathrm{m}} = 1.25$ Hz, and are more pronounced in OF than



in the experiment. The amplitude of rotor speed and torque oscillations is higher than in below-rated wind; the controlled wind turbine is more sensitive to wind speed fluctuations than in below-rated wind, and this is due to the combination of different

aerodynamic behavior of the rotor and the action of the pitch controller. As in below-rated wind, the amplitude of rotor speed oscillations is proportional to $\Delta U$; however, contrarily than in below-rated, it is higher in OpenFAST than in the experiment. Blade pitch variation is proportional to rotor speed oscillations; the amplitude is higher in OpenFAST than in the experiment and phase shift with respect to motion is different.

The thrust response and the torque response show the same phase shift with respect to platform motion, it is different than

$180°$, and is in opposition of phase with respect to blade pitch. This suggests the aerodynamic response of the controlled wind turbine is driven by blade pitch more than platform motion. OpenFAST simulations are repeated with ideal pitch actuator (i.e., $G_{\mathrm{act}}(s) = 1$), but results are very similar to those with the pitch actuator model and are omitted. The peak-to-peak amplitude of thrust and torque oscillations in the experiment is lower than in OpenFAST, but the difference is smaller than for below-rated wind. Models with dynamic polars give similar results of models with static polars. The amplitude of all quantities is lower with

dynamic BEM than with static BEM and the former is closer to the experiment. This is in agreement with the results of Bergua et al. (2022) where simulations are carried out with several codes with prescribed surge motion and prescribed harmonic blade pitch of $1.5°$ amplitude. Fast changes of blade pitch angle are known to cause dynamic inflow effects and large dynamic loads (Snel and Schepers (1995)). However, in the case of harmonic platform motion with active blade pitch control, the OpenFAST solution with dynamic inflow model results in smaller peak-to-peak variations that the one with steady inflow; this results is

similar to what is found by Berger et al. (2022) that has analyzed the dynamic inflow effect due to coherent sinusoidal wind field. Interestingly, aerodynamic loads in the experiment are even lower than in the dynamic BEM models.

The aerodynamic thrust and torque response to apparent wind is computed from results of the experiment and OpenFAST simulations (dynamic BEM and unsteady airfoil polars) with the linearized model of Sect. 4.4 and is shown in Fig. 10. To compare the below-rated and above-rated conditions, the amplitude of rotor loads ($\Delta F_x$ and $\Delta M_x$) is normalized according to

the amplitude of hub motion $A_{\mathrm{hub}}$. Linear regression based on experimental and numerical data is also computed.

Aerodynamic rotor loads for below-rated wind are linearly proportional to reduced frequency of motion, and so to the rotor apparent wind. The linear regression gives zero variation of the aerodynamic loads at $f_r = 0$ (i.e., with no platform motion). This also confirms that loads components due to rotor speed variation, that have been subtracted from total aerodynamic forces based on Eq. 28-29 are linearly proportional to rotor speed (collective pitch is constant in below-rated wind) and are correctly

captured by the linearized aerodynamic model. The agreement between OpenFAST and the experiment is in general good; the amplitude of $\Delta T$ is very close in OF and the experiment, the amplitude of $\Delta Q$ in OF is slightly higher than in the experiment. This difference may be related to the difference in the $K_{UQ}$ sensitivity already seen in Fig. 7. The phase shift is close to $-90°$; there is some dispersion for experimental torque data at higher frequencies; the phase shift predicted by OpenFAST is $-96°$.

In above rated wind, the turbine is controlled with collective blade pitch. Variations of rotor loads due to rotor speed and

blade pitch are subtracted from aerodynamic forces to isolate the contribution due to apparent wind. The amplitude of loads obtained in the experiment and in OpenFAST is aligned to the linear fit, aerodynamic forces are linearly proportional to rotor speed oscillations and blade pitch variations. The amplitude of thrust oscillations due to apparent wind in the linearized



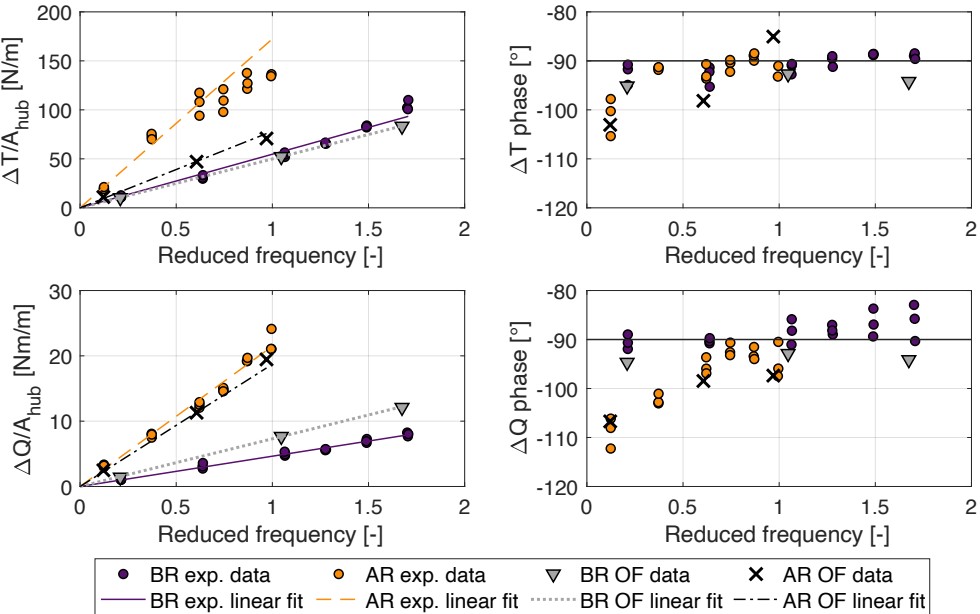

**Figure 10.** Normalized aerodynamic rotor thrust ($\Delta T/A_\mathrm{hub}$) and torque ($\Delta Q/A_\mathrm{hub}$) variation and phase shift with respect to platform pitch motion during unsteady wind in below rated (BR) and above rated (AR) conditions, for the experiment (exp.) and OpenFAST (OF).

model is lower in OpenFAST than in the experiment; thrust oscillations in Fig.9 have similar amplitude in OpenFAST and the experiment, thus the discrepancy seen in Fig. 10 may be due to differences in the aerodynamic sensitivities. The amplitude of

thrust oscillations due to apparent wind is instead the same for experimental and OpenFAST results, meaning the blade pitch and rotor speed contributions to torque variations are estimated with the same accuracy. The phase shift of thrust and torque is between $-110°$ and $-95°$, it rises linearly with frequency up to $f_\mathrm{r} = 0.75$ and is constant above. The trend of phase shift is similar for OpenFAST and experimental data, and the agreement is better for torque than for thrust. Since the agreement for the phase shift of loads due to apparent wind is good, the difference in phase shift seen in Fig. 10 is attributed to blade pitch

actuation.

## 6 Conclusions

This article presented a wind tunnel experiment whose aim is to investigate the aerodynamic response of a floating wind turbine subjected to platform pitch motion and with active control functionalities. A theoretical framework is proposed to downscale the reference open-source controller ROSCO and use it to control a scaled version of the IEA 15 MW turbine. The

controller preserves the algorithms of its full-scale version, but is run in real-time at model scale to respect the time scale of the experiment. Due to this choice, the controller parameters were downscaled; the scaling procedure is model based, and uses





information about the aerodynamic response and inertial properties of the scale model to reproduce the rotor response of the full-scale turbine.

Testing is conducted with fixed foundation and with prescribed platform pitch motion measuring the rotor response. The
experiment is modeled in OpenFAST and results of simulations are compared to those of testing. The rotor speed-blade pitch-thrust-torque characteristics of the scaled turbine are correctly captured by the OpenFAST model and are representative of the IEA 15 MW. The largest differences are seen for torque and are attributed to an offset in blades pitch that occurred in the experiment, and to uncertainty of the drivetrain efficiency.

With platform pitch motion, the turbine response is different in below-rated and above-rated wind, depending on the turbine
control strategy. Below rated, blade pitch is saturated, TSR is regulated acting on generator torque with small oscillations of rotor speed (the maximum peak-to-peak amplitude is 2% of mean value) due to the apparent wind created by platform motion. The aerodynamic response is linearly proportional to rotor-apparent wind and follows the quasi-static theory. Amplitude of thrust oscillations is correctly predicted in OpenFAST, whereas torque oscillations in the experiment are of lower amplitude than in simulations. Above rated, generator speed is regulated with collective blade pitch. The amplitude of aerodynamic
response is due to the linear combination of variations of apparent wind, rotor speed, and blade pitch. However, the phase does not follow the quasi-steady theory. The amplitude of blade pitch and aerodynamic loads is different in OpenFAST and in the experiment. Slightly better agreement is obtained with the use of dynamic BEM. Phase of blade pitch variations in OpenFAST is different than in the experiment. This can be either due to the scale model blade-pitch actuators behaving differently than expected, or modeling of the coupled aero-servo-dynamic response in OpenFAST.

In conclusion, this work has provided guidance on how to include reference wind turbine control functionalities in scale model testing of floating wind turbines. It has also confirmed the aerodynamic load response with platform motion and active control is more difficult to model than when rotor speed and blade pitch are fixed. Above rated wind, where the turbine is controlled with collective blade pitch, the turbine response is not quasi-steady and is more difficult to predict.

In tuture work, more codes, possibly of higher fidelity, can be used to model the wind tunnel experiment. The present
research examined the case of active turbine control and prescribed motion in one DOF. In future, the turbine aerodynamic response should be examined in case of realistic platform motion due to wind-wave excitation and active turbine control.

*Data availability.*  The OpenFAST model, the MATLAB Simulink version of the reference open-source controller ROSCO, and experimental data can be downloaded at Fontanella et al. (2023b).



## Appendix A: Appendix A

We provide here the analytical expressions of rotor aerodynamic sensitivities. The rotor speed to rotor torque sensitivity is:

$$K_{\omega Q} = \frac{Q_0}{\omega_{\mathrm{r},0}} \left. \frac{\partial C_Q}{\partial \lambda} \right|_0 \frac{\lambda_0}{C_{Q,0}}, \tag{A1}$$

the wind speed to rotor torque sensitivity is:

$$K_{UQ} = \frac{Q_0}{U_0} \left( 2 - \left. \frac{\partial C_Q}{\partial \lambda} \right|_0 \frac{\lambda_0}{C_{Q,0}} \right), \tag{A2}$$

and the collective blade pitch angle to rotor torque sensitivity is:

$$K_{\beta Q} = \frac{1}{2} \rho \pi R^3 U_0^2 \left. \frac{\partial C_Q}{\partial \beta} \right|_0, \tag{A3}$$

where $\partial C_Q/\partial \lambda$ and $\partial C_Q/\partial \beta$ are the two components of the $C_Q$ gradient.

The rotor speed to rotor thrust sensitivity is:

$$K_{\omega T} = \frac{T_0}{\omega_0} \left. \frac{\partial C_T}{\partial \lambda} \right|_0 \frac{\lambda_0}{C_{T,0}}, \tag{A4}$$

the wind speed to rotor thrust sensitivity is:

$$K_{UT} = \frac{T_0}{U_0} \left( 2 - \left. \frac{\partial C_T}{\partial \lambda} \right|_0 \frac{\lambda_0}{C_{T,0}} \right), \tag{A5}$$

and the collective blade pitch angle to rotor thrust sensitivity is:

$$K_{\beta T} = \frac{1}{2} \rho \pi R^2 U_0^2 \left. \frac{\partial C_T}{\partial \beta} \right|_0, \tag{A6}$$

where $\partial C_T/\partial \lambda$ and $\partial C_T/\partial \beta$ are the two components of the $C_T$ gradient.

## Appendix B: Appendix B

We demonstrate that running the turbine controller at full-scale with scaling of input and output signals does not preserve the turbine frequency response.

Let us consider here the response of the wind turbine to a change in wind speed when it function in full load and rotor speed is controlled with a PI collective blade pitch controller. Assuming the wind speed rate of change is low, the variation of rotor speed is:

$$F9\overline{\omega}_{\mathrm{r}} = -\frac{K_{UQ}}{K_{\omega Q}} \overline{U}. \tag{B1}$$

This variation of rotor speed is counteracted by the PI collective pitch controller, which frequency response function for the IEA 15 MW is shown in Fig. B1.



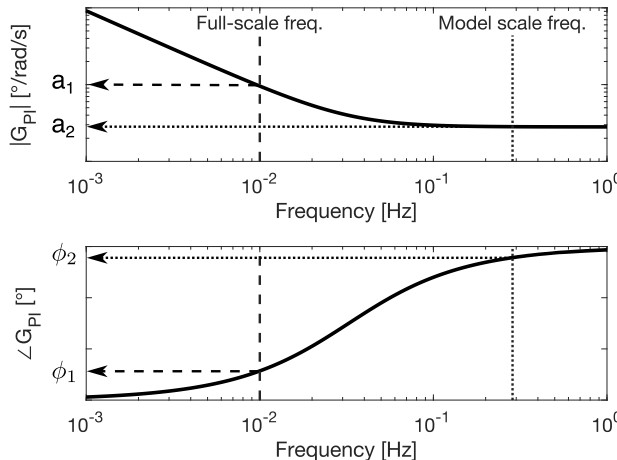

**Figure B1.** Frequency response function (amplitude and phase) of the IEA 15 MW PI collective blade pitch controller. The frequency response function evaluated at "Full-scale freq." has amplitude $a_1$ and phase $\phi_1$. If it is evaluated at "Model scale freq." it has amplitude $a_2$ and phase $\phi_2$.

We assume the variation of wind speed for the full-scale turbine is harmonic. In general, this can be due to a wind gust or due to the apparent wind created by motion in case of a floating wind turbine. The wind speed variation is:

$$\overline{U} = u\sin\left(2\pi f_{\mathrm{w}}t\right), \tag{B2}$$

where $f_{\mathrm{w}}$ is the frequency of the wind speed oscillations. Using Eq. B1, the rotor speed response due to the wind speed is:

$$\overline{\omega}_{\mathrm{r}} = -\frac{K_{UQ}}{K_{\omega Q}}\left(u\sin\left(2\pi f_{\mathrm{w}}t\right)\right). \tag{B3}$$

The blade pitch controller reaction to this oscillation is:

$$\overline{\beta} = a_1\left(-\frac{K_{UQ}}{K_{\omega Q}}\right)\left(u\sin\left(2\pi f_{\mathrm{w}}t + \phi_1\right)\right), \tag{B4}$$

where $a_1$ and $\phi_1$ are the amplitude and frequency of the PI pitch controller frequency response function at $f_{\mathrm{w}}$.

Let us consider a scaled version of the full-scale turbine. The wind speed oscillation for the model has scaled amplitude and scaled frequency (e.g., when the turbine is mounted on a scale model of the floating platform):

$$\overline{U} = (u\lambda_v)\sin\left(2\pi f_{\mathrm{w}}\lambda_f t\right), \tag{B5}$$

where $\lambda_f = \lambda_v\lambda_L^{-1}$. Assuming the rotor response of the turbine model is ideally scaled:

$$\overline{\omega}_{\mathrm{r}} = \left(-\frac{K_{UQ}}{K_{\omega Q}}\frac{1}{\lambda_L}\right)(u\lambda_v)\sin\left(2\pi f_{\mathrm{w}}\lambda_f t\right) \tag{B6}$$

When the turbine controller is operated in real-time in full-scale mode, inputs from the turbine model are scaled up to full-scale values before going into the controller. For rotor speed:

$$\overline{\omega}_{\mathrm{r,fs}} = \left(\left(-\frac{K_{UQ}}{K_{\omega Q}}\frac{1}{\lambda_L}\right)(u\lambda_v)\frac{1}{\lambda_f}\right)\sin\left(2\pi f_{\mathrm{w}}\lambda_f t\right), \tag{B7}$$





where $\lambda_v \lambda_L^{-1} \lambda_f^{-1} = 1$. The blade pitch controller response is:

$$\overline{\beta} = a_2 \left( -\frac{K_{UQ}}{K_{\omega Q}} \right) \left( u \sin \left( 2\pi f_{\mathrm{w}} \lambda_f t + \phi_2 \right) \right), \tag{B8}$$

where $a_2$ and $\phi_2$ are the amplitude and frequency of the PI pitch controller frequency response function at $f_{\mathrm{w}} \lambda_f$. $\overline{\beta}$ is non dimensional and, with the full-scale controller approach, it is applied to the turbine scale model without any further operation.

Comparing Eq. B4 to Eq. B8 we see they have different amplitude and phase, that are due to the frequency response function of the PI blade pitch controller rather than dimensional scaling.

*Author contributions.*   AF and ED devised the methodology to downscale the wind turbine controller and created the simulation model. FN, AF and MB designed the wind tunnel experiment, that was run by AF, ED and FN. ED and AF analyzed the experimental data and run the numerical simulations. MB supervised the research project. All coauthors thoroughly reviewed the article.

*Competing interests.*   The authors declare that they have no conflict of interest.

*Acknowledgements.*   The project has received funding from the European Union's Horizon 2020 research and innovation program under
grant agreement No. 860737 (STEP4WIND project, STEP4WIND).



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
