# Peer review of "Controller design for model-scale rotors and numerical-experimental study using prescribed motion"

_Wind Energy Science, 2023_

## Referee Comment (RC2)

[revised manuscript text omitted]

*writing style need to be improved.
too many imprecisions in the description without clear context, e.g. reference value, simulation results, theoretical estimate.
The paper is kind of a long washing list of observations between simulations and experiments,*

Need to structure the discussions of the results in a logical manner so that the reader can follow.

The many observed differences often are not really relevant for the main objectives of the paper.

Need to focus really on discrepancies that are relevant and offer clearer explanations to the potential discrepancies

Controller scaling procedures are rather unstructured and difficult to follow.

[revised manuscript text omitted]

---

## Author Comment (AC1)

Politecnico di Milano
Department of Mechanical Engineering
Via La Masa 1, 20156, Milan
Italy

Wind Energy Science Discussion

Date: June 21, 2023
Subject: WES-2023-39 Final Response

Dear Referees,

We would like to thank you for having reviewed our manuscript and for the valuable feedbacks. Your suggestions focused our attention on aspects we didn't consider in the first version of the article, and we believe this will improve the quality and impact of this work.

The article has been revised following your suggestions. We hope to have addressed most of your concerns which helped us improve the article, otherwise we are happy to continue the discussion.

On behalf of all Authors,
yours sincerely,

Alessandro Fontanella

Attached documents:
- Response to Anonymous Referee #1
- Response to Anonymous Referee #2
- List of major changes
- Latexdiff

**Response to Anonymous Referee #1**

Dear Referee,

Thank you for taking the time to review our manuscript and for the valuable comments you made. We appreciate your suggestion for the title, and we would like to use it because it summarizes better our work. Thank you for reminding us of hybrid wave basin testing which also faced and solved the problem of including the controller in experiments about floating wind turbines.

Below you can find our answers to your comments.

| RC1.1 | We suggest to a clarify the title, because "with inclusion of reference control functionalities" was not clear to us to start with. We suggest the somewhat lengthy title: "Controller design for model-scale rotors, and numerical/experimental study using prescribed motions". |
|---|---|
| AC1.1 | We think your title conveys the message of the article better than the one we used in the first version of the manuscript, thus we decided to adopt it. |

| RC1.2 | Abstract: the statement "Aerodynamic loads calculation in these tools has been recently validated..." should be completed by "for low-frequency motions", as indicated in the introduction. This is important to pinpoint when dealing with floating wind turbines, experiencing large wave-frequency motions. |
|---|---|
| AC1.2 | We agree with this comment, and we have pointed out across the text that the type of motion we are considering is of low frequency. |

| RC1.3 | In the abstract, and more generally in the paper, it should be emphasized that only thrust and torque loads have been investigated in the present study. Some of the other four components of the aerodynamic loading are important for the response of some floater concepts. See for instance Bachynski et al. (2015) https://doi.org/10.1016/j.egypro.2015.11.400 |
|---|---|
| AC1.3 | We included this comment in the conclusion of the article, saying future work should also address the other components of rotor loads. |

| RC1.4 | In the introduction it is stated that "In the last decade, several scale model experiments about the wind-wave response of floating wind turbines have been carried out, and a review of them is presented by Gueydon et al. (2020). The large majority of tests involving a scaled wind turbine did not use active turbine control.". This statement is inaccurate.
A large number (probably the majority) of today's concepts supporting 5MW-15MW turbines have been tested in wave tanks using a hybrid (cyber-physical) approach, at SINTEF, Marin, UHC, etc... with literature references easy to find. This approach models the actual full-scale wind-turbine controller. Even though the present work is of course targeting tests using physical wind (and a "performance-matching" rotor), it should be emphasized that this approach is not the only one, and particularly that the hybrid approach solves the issue related to the downscaling of the controller. |
|---|---|

[Figure]

| AC1.4 | We agree about the importance of hybrid wave basin testing, and we think this methodology must be mentioned in the literature review because it is the dual of what we discuss. We have added a paragraph in the introduction describing few hybrid experiments recently carried out in different laboratories. |
|---|---|

| RC1.5 | Related to the previous point, in the introduction: "The methodology we developed to integrate active control in experiments and simulations should benefit future scale model testing activities" should be completed by "using physical wind". |
|---|---|
| AC1.5 | We pointed out the methodology we propose is meaningful to experiments that have physical wind and a wind turbine scale model. |

| RC1.6 | In section 2.1, it is indicated that the tower eigenfrequency associated to the fore-aft mode is 9.5 Hz. Can the authors elaborate on how this corresponds to the full-scale frequency of current designs? Are the aerodynamic thrust and torque investigated here affected by possible difference in vibrations frequencies between current designs and the model used here? |
|---|---|
| AC1.6 | We added the full-scale value of our tower eigenfrequency. In our approach, we neglect the variations of aerodynamic loads due to tower vibrations. We explained that in Section 4.2 when we describe the estimation of aerodynamic loads. |

| RC1.7 | Line 216, it is stated that "loads measured by the load cell are mostly due to inertia". The varying projection of the acceleration of gravity on the horizontal accelerometer is likely to be important too when the rotor is undergoing pitch motions? |
|---|---|
| AC1.7 | Yes, you are right, and we pointed out there is a force contribution due to gravity. |

| RC1.8 | To conclude with a minor comment: Typo "fro" line 182. |
|---|---|
| AC1.8 | Done. |

**Response to Anonymous Referee #2**

Dear Referee,

We would like to thank you for the accurate feedback. We agree with you about the first version of the manuscript being not clear concerning the controller design procedure, but mostly, it is true it was difficult to understand how to use the results of this study and their possible impact on future work.

Following your suggestion, we made some major changes to the article. We reworked the section about the scaling of the wind turbine controller (Sect. 3.1), and we simplified the notation used in the equations. We preferred to remove the quasi-steady steady model of rotor aerodynamic loads (was Sect. 4.4): the model derivation was a bit difficult to follow and the results obtained by means of it were not so useful compared to the rest of the article. In the analysis of the wind turbine response with platform pitch motion, we decided to focus on one motion condition, instead of three, choosing the one with the largest differences between numerical simulations and experiment. To clarify the differences between numerical simulation and experiment, we added a new set of results that we obtained running simulations with the stand-alone aerodynamic model of OpenFAST. Finally, we changed the conclusions and the abstract to better fit the reviewed article.

| RC2.1 | The writing style and clarity of the writing need to be improved significantly. The subject of the manuscript is not easy to follow given the way it is explained in the paper. It would be beneficial to include a table of symbols due to the many symbols used in the derivation of the scaled controller, tuning, and modifications. |
|---|---|
| AC2.1 | We revised the article to make it easier to follow. We added a list of the symbols that occur most often in the text. |

| RC2.2 | The procedure how the controller is designed, scaled, and tuned is difficult to follow. It would be easier to simplify and remove some of the details that are not relevant to the paper. |
|---|---|
| AC2.2 | After having revised the article, we agree with you that the controller design procedure of the first version of the manuscript was hard to follow. We simplified this part of the text, and we changed the notation of the equations.
Please let us know if you think it could be improved further. |

| RC2.3 | The manuscript used often unprecise languages and words without giving a clear context, for example "reference value", the reader often struggles to understand what exactly the "reference value" is referring to. The same with terms like simulation results or estimates etc. where no clear context is given to help reader understand which simulation results the authors are referring to. Another example is the shortening of the terms, while clear for many, it is better not to speak about "below rated wind" instead of the complete term "below rated wind speed". This kind of shortening of terms gives the reader the impression of lack of precision in the writing. |
|---|---|

| AC2.3 | Thank you for this suggestion. We have thoroughly revised the text to make it easier to understand. |
|---|---|

| RC2.4 | The manuscript gives the impression that when the authors describe the results of the experiment and the simulations, it is presented in a kind of a very long laundry list without context and logical connection. The readers are overwhelmed with a lot of information without knowing which ones are actually relevant or are important. It would be good to restructure the results and discussion in a more logical way, and possibly discard observations that, while interesting, have little practical or theoretical value. Instead expand the discuss the discrepancies in more details and explanations of possible causes of the discrepancies and the consequence when using the results of the study. |
|---|---|
| AC2.4 | We agree the results section of the first version of the manuscript was a list of the main findings and it was not clear how to use them. Following your suggestion, we restructured the results section in this way:

• At the beginning of Section 5, we briefly explain how results are structured and which is their purpose.
• In the results obtained with fixed turbine, we removed unnecessary details, and we focussed on the main differences between experiment and simulation, explaining their reason, the impact on the implementation of closed-loop controls, and possible strategies to mitigate these discrepancies.
• For results with platform pitch motion, we decided to focus on just one condition among the three that were presented in the original manuscript. We selected the condition with the largest apparent wind and with the most significant differences between simulation and experiment. We added a new set of results about the aerodynamic loads computed in OpenFAST that clarify the role of the aerodynamic model in the simulation of the wind turbine scale mode with closed-loop control.
• We decided to remove results obtained with the linearized model of aerodynamic thrust and torque (lines 392-415 of the first version of the manuscript). Even if this kind of modelling is used in the study of the unsteady aerodynamic response of wind turbines, here it was disjointed from the rests of the results and it was not so important in the discussion of the implementation of the closed-loop controller. Because of this choice, we also removed the derivation of the linearized thrust and torque (was Sect. 4.4). |

| RC2.5 | the manuscripts contain spelling mistakes and more specific comments can be found in the attached PDF file. |
|---|---|
| AC2.5 | We revised the text according to your comments to fix typos. |

**List of major changes**

We provide here a list of the main modifications.

- Results:
  a. At the beginning of Section 5, we briefly explain how results are structured and which is their purpose.
  b. In the results obtained with fixed turbine, we removed unnecessary details, and we focussed on the main differences between experiment and simulation, explaining their reason, the impact on the implementation of closed-loop controller, and possible strategies to mitigate these discrepancies.
  c. For results with platform pitch motion, we decided to focus on just one condition among the three that were presented in the manuscript. We selected the condition with the largest apparent wind, and with the largest differences between simulation and experiment. We added a new set of results about the aerodynamic loads computed in OpenFAST that clarify the role of the aerodynamic model in the simulation of the wind turbine scale mode with closed-loop control.
- Conclusions: we modified the conclusions section to be aligned to the new results section.
- Abstract: the abstract has been modified clarify the goals of the article and summarize its main results.
- Removed linearized model:
  a. We decided to remove results obtained with the linearized model of aerodynamic thrust and torque (lines 392-415 of the first version of the manuscript). Even if this kind of modelling is used in the study of the unsteady aerodynamic response of wind turbines, here it was disjointed from the rests of the results and it was not so important in the discussion of the implementation of the closed-loop controller.
  b. We also removed the derivation of the linearized thrust and torque (was Sect. 4.4).
- List of symbols: we added a table with the most used symbols in Appendix C.
- Figure 6: the OpenFAST results shown in the previous version were incorrect.
- Figure 7: plots have been arranged in two rows instead of two columns to save space, but nothing has changed in the results shown in the plot.

**Latexdiff**

[revised manuscript text omitted]
_{des,2}(U_0)\Omega_{0,fs} = \sqrt{\frac{\tau_g K_{\beta Q,fs}(U_0) \, k_{I,\beta,fs}}{J_{fs}^*}} \left(\sqrt{\frac{\tau_g K_{\beta Q} \, k_{I,\beta}}{J^*}}\right)_{0,fs}, \tag{12}$$

$$h_{\underline{\text{des}},2}(U_0)_{0,\text{fs}} = \frac{\tau_g K_{\beta Q,\text{fs}}(U_0) k_{\text{P},\beta,\text{fs}} + K_{\omega Q,\text{fs}}(U_0)}{2 J_{\text{fs}}^* \, \omega_{\text{des},2}(U_0)} \left( \frac{\tau_g K_{\beta Q} k_{\text{P},\beta} + K_{\omega Q}}{2 J^* \Omega} \right)_{0,\text{fs}}, \tag{13}$$

where  $(\cdot)_{0,\text{fs}}$ denotes full-scale quantities. , that are evaluated at the operating point identified by $U_0$ in case of wind speed-dependent values;

230  2. the closed-loop frequency and damping of the WTM are computed by scaling dimensionally those of the IEA 15 MW:

$$k_{\text{I},\beta}(U_0 \Omega_{0,\text{sm}} = \Omega_{0,\text{fs}} \cdot \lambda_v) = \frac{J_{\text{sm}}^* \left( \
[revised manuscript text omitted]

$$\Delta Q(t) = -K_{UQ}(2\pi f_{\text{m}})A_{\text{hub}}\cos(2\pi f_{\text{m}} t)\,,$$

with $A_{\text{hub}} = A_{\text{m}} d_{\text{hub}}$, where $d_{\text{hub}}$ is the hub distance from the center of platform pitch rotation ($d_{\text{hub}} = 1.48$ m, see Fig. 1). When the turbine aerodynamic response is described by Eq. ??-??, the phase of the force response with respect to motion is
360 $-\pi/2$, and the zero-peak amplitude normalized by amplitude of hub motionis linear with motion frequency:

$$\Delta T/A_{\text{hub}} = 2\pi f_{\text{m}} K_{UT}\,,$$

$$\Delta Q/A_{\text{hub}} = 2\pi f_{\text{m}} K_{UQ}\,.$$

[revised manuscript text omitted]

---

## Author Response (AR2)

Politecnico di Milano
Department of Mechanical Engineering
Via La Masa 1, 20156, Milan
Italy

Wind Energy Science

Date: August 3, 2023
Subject: WES-2023-39 Author's Response

Dear Referees and Editors,

We would like to thank you for your efforts in the article review process.

Attached to this letter you can find a point-by-point response to the Reviewer comments.

On behalf of all Authors,
yours sincerely,

Alessandro Fontanella

**Response to Anonymous Referee #3**

| RC3.1 | The title of the manuscript has been changed to: "Controller design for model-scale rotors and numerical-experimental study using prescribed motion". It might be better to change it further to: "Controller design for model-scale rotors and validation using prescribed motion". |
|---|---|
| AC3.1 | We changed the title according to your suggestion. |

| RC3.2 | Fig 8-9 do not show experimental results below-rated power without control. Why? |
|---|---|
| AC3.2 | For the below-rated wind speed, the experimental results with fixed rotor speed/blade pitch are omitted, because when the WTM is controlled with an open-loop rotor speed set point rotor speed is not fixed but has oscillations |

| RC3.3 | Typos |
|---|---|
| AC3.3 | Fixed. |